# Population Genetic Structure Analysis Reveals Decreased but Moderate Diversity for the Oriental Fire-Bellied Toad Introduced to Beijing after 90 Years of Independent Evolution

**DOI:** 10.3390/ani11051429

**Published:** 2021-05-17

**Authors:** Yang Teng, Jing Yang, Guofen Zhu, Fuli Gao, Yingying Han, Weidong Bao

**Affiliations:** 1College of Biological Sciences and Technology, Beijing Forestry University, Beijing 100083, China; tengyang970311@163.com (Y.T.); fuligao@bjfu.edu.cn (F.G.); thinkinghyy@bjfu.edu.cn (Y.H.); 2Institute of Zoology, Chinese Academy of Sciences, Beijing 100084, China; yangjing201777@163.com; 3College of Biology and Food Science, Hebei Normal University for Nationalities, Chengde 067000, China; princejks@sina.com

**Keywords:** *Bombina orientalis*, mitochondrial DNA, microsatellite marker, independent evolution, genetic variation

## Abstract

**Simple Summary:**

Habitat isolation and loss are significant factors that lead to the decline of wildlife populations worldwide, and habitat loss further leads to the shrinkage of populations, which increases the risk of inbreeding and the genetic decline of the populations. To explore the independent evolutionary characteristics of different populations, this study analyzed the genetic disparity of the introduced oriental fire-bellied toad in Beijing from a source population in Shandong Province. The results show that, despite originating from a small artificially introduced population, the toads in the Beijing region have maintained a moderate genetic diversity after 90 years of independent evolution, indicating that this species has a high capacity for survival and adaptation.

**Abstract:**

Detailed molecular genetic research on amphibian populations has a significant role in understanding the genetic adaptability to local environments. The oriental fire-bellied toads (*Bombina orientalis*) were artificially introduced to Beijing from Shandong Province in 1927, and since then, this separated population went through an independent evolution. To explore the differentiation of the introduced population with its original population, this study analyzed the genetic structure of the oriental fire-bellied toad, based on the mitochondrial genome control region and six microsatellite sites. The results showed that the haplotype diversity and nucleotide diversity of the mitochondrial D-loop region partial sequences of the Beijing Botanical Garden population and the Baiwangshan population were lower than those of the Shangdong Kunyushan population. Microsatellite marker analysis also showed that the observed heterozygosity and expected heterozygosity of the Beijing populations were lower than those of the Kunyushan population. The phylogenetic trees and network diagrams of haplotypes indicated that the three populations were not genetically separated. However, the structure analysis showed a genetic differentiation and categorized the sampling individuals into Beijing and Shandong genetic clusters, which indicated a tendency for isolated evolution in the Beijing population. Although the Beijing populations showed a decline in genetic diversity, it was still at a moderate level, which could maintain the survival of the population. Thus, there is no need to reintroduce new individuals from the Kunyushan source population.

## 1. Introduction

Habitat isolation and loss are significant factors that lead to the decline of wildlife populations worldwide, and habitat loss further leads to the shrinkage of populations, which increases the risk of inbreeding and the genetic decline of populations [1]. Genetic diversity, as an indicator evaluating the living status and viability of a population, plays an important role in assessing population trends and species conservation. Due to their relatively low dispersal capacity, amphibians are good representatives for studying the genetic differentiation of isolated geographical populations [2]. In Europe, *Bombina variegata* and *Bombina bombina* have been widely studied in the area of molecular biology [1], such as through population structure analysis using mitochondrial control sequences and the polymorphism of microsatellite sites for *Bombina bombina*, revealing obvious genetic differences between sampling populations due to habitat fragmentation. A small-scale multilocus phylogeography analysis on *Bombina variegata* and *Bombina bombina* indicated that no genetic exchange has taken place between the two species [3].

Five species of Bombinatoridae are found in China, and three species are deemed to be threatened [4]. Habitat loss, wetland pollution, and over-exploitation for medicine and food are the main factors causing the reduction in the populations of these amphibians [5]. For the species under the genus of *Bombina*, including *Bombina orientalis*, *Bombina fortinuptialis*, and *Bombina maxima*, distributed within the territory of China, there has been little research work conducted on their genetic structure. A study on the mitochondrial structure analysis of *Bombina orientalis* found population genetic differentiation as a result of isolated evolution for geological change in northeast China [6]. The evolutionary history study of four lineages of *Bombina orientalis* indicated that the genetic structure diversity was caused by historical geological activities [7].

The existing oriental fire-bellied toad is the only species of the *Bombina* genus living in northern China and is mainly distributed in Hebei, Heilongjiang, Liaoning, Jilin, Inner Mongolia, Anhui, Shandong, and Beijing; it is also distributed in peninsular Korea and the far east of Russia [6]. In 1927, Mr. Liu Cheng-chao collected and brought more than 200 *B. orientalis* individuals from Kunyushan in Yantai to Beijing and released them in the Yingtaogou of the Xiangshan Wafo Temple and in the wetland near the Western Mountain, where they have since reproduced [8]. Currently, research on *B. orientalis* mainly focuses on its baseline biological characteristics, reproductive ecology, population status, and food composition [7,9]. It is commonly used in behavioral, ecological, endocrinological, and histological studies as a laboratory animal [10,11]. At present, studies based on mitochondrial gene molecular markers regard the Shandong and Beijing populations as the same cluster branch [9,12]. In our preliminary study, however, we found that there were genetic differentiations of the two geographical populations.

Under the neutral model, a population’s genetic diversity depends on its effective population size and gene mutation rate; genetic diversity will increase with a larger effective population size and the decreasing effects of drift. Higher population genetic diversity in the abundant species is likely due to a combination factors [13]. The goal of this study was to measure the genetic diversity of *B. orientalis* in Beijing that differ from the source population in size after long time independent evolution, and to see whether genetic diversity is related to population size or not. In order to detect the actual situation and explore the detailed independent evolutionary trajectory of this species in Beijing, we analyzed the genetic structure diversity using mitochondrial and microsatellite markers based on samples from two areas in Beijing and from the Kunyushan source population in Shangdong Province.

## 2. Materials and Methods

### 2.1. Sampling Areas

Samples from Beijing were collected at the Beijing Xiangshan Botanical Garden (ZWY) in 2017 and Baiwangshan (BWS) Forest Park in 2018. Samples from Shandong Province were collected from Kunyushan (KYS) of Yantai city in 2017. A total of 63 samples were collected from the three sampling locations (26 from the Beijing Botanical Garden, numbered ZWY1-ZWY26; 8 from Baiwangshan, numbered BWS1-BWS8; and 29 from Kunyushan, numbered KYS1-KYS29). The method of euthanizing was to use diethyl ether deep anesthesia. The individuals were adults with body lengths between 40 and 45 mm. The hind leg muscles were taken off and fixed with anhydrous ethanol for sampling in a centrifuge tube, then stored in a refrigerator at −20 °C for DNA extraction. The treatment of the animals followed the general protocol for experimental animal processing (there was no experimental animal ethics management unit in our institution when this study carried out).

### 2.2. DNA Extraction and Mitochondrial Control Region Sequence Amplification

Total DNA was extracted using the TIANamp Genomic DNA Kit (Tiangen Biotech Co. Ltd., Beijing, China). The primers were designed referring to the D-loop segment in the complete mitochondrial DNA sequence of *Bombina orientalis* (GenBank accession number AY585338) and synthesized by Ruibio Biotech Co., Ltd. (Beijing, China). Three pairs of primers were used to amplify the mitochondrial D-loop sequence: DFLC-1 (F: 5′-TAGAGATTTGCTATGCTTGT-3′, R: 5′-GGCGTATGGGTTTTTAAAAT-3′), DFLC-2 (F: 5′-GTGTCCAGGATCACCAACTT-3′, R: GGGCTCATCTCAGCATCTTC), and DFLC-3 (F: 5′-ATAAACGTAAAATAGAGCC-3′, R: 5′-ATAGATTCACATCCGTCA-3′). Different primers had overlapping fragments of at least 100 bp in the mitochondrial genome to improve the sequencing accuracy.

The PCR amplifications were conducted in a total volume of 20 μL, including 10 μL of Extaq enzyme (Takara, Shiga, Japan), 0.8 μL of each forward and reverse primer (10μmol/L), 2 μL of DNA, and DNase/RNase-free deionized water. The PCR program was as follows: pre-denaturing for 4 min at 95 °C, denaturing for 40 s at 95 °C, annealing for 40 s at 50 °C or 52 °C, extension for 40 s at 72 °C, followed by 40 cycles; this was extended to 72 °C for another 7 min. The PCR products were detected by 1% agarose gel electrophoresis, and sequencing was performed by Ruibio Biotech Co., Ltd. (Beijing, China) using the ABI3730XL sequencer.

The sequencing results were compiled in the Contig Express software and complemented by manual checking. Sequence alignment was performed by the ClustalX 2.1 software [14], and the non-target fragment sequence and the 3′-end repeat sequence of the D-loop region were removed.

### 2.3. Primer Selection and Amplification of Microsatellite Sites

The primers of 25 microsatellite sites of *B. bombina* and *B. variegata* based on the reports of Guicking [15] and Nümberger [16] were selected to be amplified with suitable microsatellite site primers in this study. After optimizing the PCR conditions, 6 pairs of primers with a better amplification were selected; these were 9H, B14, B13, 8A, 12F, BV11.7 [17]. A fluorescent label was added to the 5′ end of each forward primer for further amplifications.

The PCR reaction system was the same as the amplification reaction of the mitochondrial control region. The PCR program was as follows: pre-denaturing for 4 min at 95 °C, denaturing for 40 s at 95 °C, annealing for 30 s at 48~52 °C, extension for 30 s at 72 °C, followed by 40 cycles; this was extended to 72 °C for another 8 min. The PCR-amplified samples were sent to Norse Genomics Research Center Co., Ltd. (Beijing, China), for gene scanning in the ABI-PRISM310 genotyping machine, and the microsatellite allele band size was determined by the GeneMarker v2.4.0 software [18].

### 2.4. Data Analyses

The haplotype diversity, nucleotide diversity, and nucleotide divergence between populations and the haplotype number of the three populations were calculated in the software Dnasp5 [19]. The Kimura two-parameter method [20] in MEGA 5.2 [21] was used to calculate the genetic distance, and the neighbor-joining tree was constructed among the haplotypes with the mitochondrial control region sequence of *Bombina bombina* as the outgroup (GenBank accession number: EU115993). The phylogenetic tree was constructed in PAUP 4.0 [22] based on the maximum likelihood method, and the most suitable DNA substitution model was selected based on the Akaike information criterion (AIC) by Model Test 3.7 [23] and the PAUP 4.0 software by the bootstrap method to test the confidence at the nodes 1000 times. The median-joining method was used to construct the network diagram among haplotypes in NETWORK5.0 [24].

The population genetic structure was analyzed in the software Arlequin 3.5 [25], the genetic difference among geographical populations was compared by AMOVA, and the *F*st and P values were calculated to indicate the differentiation degree among the studied populations.

The microsatellite allele number, effective allele number, expected heterozygosity, observed heterozygosity, gene richness, Nei genetic distance, and polymorphism information content were calculated by the GenALEx 6.5 software [26] and FSTAT software [27]. The genetic differentiation coefficient *F*st and its significance among populations were calculated using the Arlequin 3.5 software [25]. The genetic differentiation among populations was analyzed and verified using the Bayesian grouping model in the Structure software [28].

## 3. Results

### 3.1. Population Genetic Diversity in Mitochondrial D-Loop Sequences

Through DNA extraction, PCR amplification, sequencing, and splicing, the mitochondrial D-loop fragment sequences of 47 samples were successfully obtained, including 24 from the botanical garden, 7 from Baiwangshan, and 16 from Kunyushan. In the 1122 bp sequence, there were 23 nucleotide polymorphism sites, 17 monomorphism sites, 6 simple information sites, and 14 haplotypes, which were defined accordingly (NCBI Serial number: MW535554–MW535567). One haplotype (BWS–ZWY) was shared by the populations of Baiwangshan and the Botanical Garden, the haplotype BWS–KYS was shared between the Baiwangshan and Kunyushan populations, and the haplotype ZWY–KYS was shared by the populations of the Botanical Garden and Kunyushan. The Botanical Garden and Baiwangshan populations had the unique haplotypes of ZWY1 and BWS1, while the population in Kunyushan had nine unique haplotypes (KYS1–KYS9).

Based on the calculation of Dnasp 5.2, the overall genetic diversity indexes of the Beijing populations were obviously lower than those of the Shandong population (Table 1). The nucleotide diversity between the Botanical Garden and Kunyushan populations, the Baiwangshan and Kunyushan populations, the Baiwangshan and Botanical Garden populations, and the Beijing populations in total and Shandong population was 0.296%, 0.226%, 0.182%, and 0.280%, respectively.

The genetic distance between the 14 haplotypes was between 0.001 and 0.011, with an average of 0.002, according to MEGA 5.2 software calculation, indicating that there was a low genetic difference among the three populations.

### 3.2. Population Genetic Differentiation on Mitochondrial Markers

AMOVA analysis showed that 18% of the variation occurred between populations and 82% occurred within populations (*p* < 0.01). The *F*st between the Beijing and Shandong populations was 0.124 (*p* < 0.01). Based on the AIC standard, the most appropriate nucleotide substitution model was the HKY+I model. The NJ tree and ML tree based on partial sequences of the mitochondrial D-loop region showed that the haplotypes between the Beijing and Shandong populations had not yet formed a significant lineage structure (Figure 1). The haplotypes were cross distributed between the two populations, and the confidence level at multiple branch nodes was low.

The network diagram indicated that there were more derived haplotypes around the KYS6 and BWS–KYS haplotypes, followed by the KYS4, ZWY–KYS, and BWS–ZWY haplotypes (Figure 2). BWS–ZWY had the highest frequency of shared haplotypes, followed by ZWY–KYS, and the individuals sharing the two haplotypes primarily originated from the ZWY population.

### 3.3. Population Genetic Diversity Based on Microsatellite Markers

Six pairs of microsatellite primers were selected with a high amplification specificity, good polymorphism, and high amplification efficiency; 60 alleles were successfully amplified from the sampling individuals, including 14 alleles that were specific to the Beijing population, 27 alleles that were specific to the Shandong population (Table 2), and 19 alleles that were shared by the Beijing and Shandong populations. The average allele number of the Beijing and Shandong populations was 7.500 and 9.667, and the number of effective alleles was 3.269 and 6.595, respectively.

The *F*st between the two populations was 0.076 (*p* < 0.01), indicating that there was a moderate degree of genetic differentiation between the two populations based on microsatellite markers. Based on the Structure software calculation, when K = 2, the average value of ln-likelihood was higher (when k = 1, ln likelihood = −1369.3. When k = 2, ln likelihood = −1235.4. When k = 3, ln likelihood = −1315). Therefore, this result supported dividing the sampling *B. orientalis* individuals into two populations (genetic clusters) of Beijing and Shandong (Figure 3).

## 4. Discussion

### 4.1. Suitability of Microsatellite Primers

In genetic studies on toads, microsatellite markers as a supplement to mitochondrial control region markers can make up for the problem that mitochondrial markers cannot reveal the genetic structure and variation level of the targeted population when the study sample size is small [29]. The microsatellite primers of toads are more difficult to commonly use between species [30], such as the eight microsatellite primers designed for *B. bombina*, which have not been successfully amplified in other toads, except being partially applied to *B. orientalis* and *B. variegata* [31]. Furthermore, the polymorphic states of the primers were quite different across species, so they still need to be filtered when conducting a genetic diversity analysis on different toad species. Similarly, in the study of Stuckas [31], only eight pairs of the 40 newly designed primers could be stably amplified a single band in *B. bombina*. Therefore, the number of stably used microsatellite primers is lower, making it difficult to study population genetics structure in the Anura species. The six pairs of microsatellite primers selected in this study were all able to be amplified stably and had a good polymorphism, which can be further used for *B. orientalis.*

### 4.2. Genetic Diversity of Bombina Orientalis in Beijing

Compared to the study based on the mitochondrial analysis of 523 individuals of the *B. orientalis* in the northeast of China, the average haplotype diversity was 0.774 (95%CI: 0.250–0.909) and the nucleotide diversity was 0.00182 (95%CI: 0.0014-0.00246) [6]; the mitochondrial genetic diversity population in Beijing was moderate, with the mitochondrial haplotype diversity being 0.641 and the nucleotide diversity being 0.00178. Based on the microsatellite marker analysis used in this study, the observed heterozygosity was 0.529 and the expected heterozygosity was 0.571. Compared to the index for *B. bombina* (Ho = 0.353–0.658, He = 0.345–0.659) and *B. variegata* (Ho = 0.38–0.70, He = 0.37–0.59) [6], our *B. orientalis* population also showed a moderate genetic diversity. In addition, the mitochondrial haplotype diversity, nucleotide diversity, observed heterozygosity, and expected heterozygosity in microsatellites were all higher in the Shandong population (Table 1 and Table 2).

Therefore, in this study, the genetic diversity of the population in Beijing was lower than that in Shandong Province, which might have been caused by the founder effect, but the overall genetic diversity was at a medium level. It should be noted that, due to the small population size, genetic drift can easily lead to the loss of rare alleles when the population decreases, which may further increase the risk of inbreeding for the Beijing population.

### 4.3. The Relationship between Ne and Genetic Diversity

A report found that genetic diversity will increase with a larger effective population size and the decreasing effects of drift and higher population genetic diversity in the abundant species was likely due to a combination of factors [13], including larger local population sizes (and presumably effective population sizes), faster generation times and high rates of gene flow with other populations. This study would conform this point: the haplotype diversity of the Shandong population was 0.950 ± 0.036 while that of the Beijing population was 0.641 ± 0.071, and the nucleotide diversity of the Shandong population was 0.00327 while that of the Beijing population was 0.00178, indicating that the genetic diversity of the Shandong population was higher than that of the Beijing population, and the Shandong population had a larger population size than that of the Beijing population; our data were predominantly consistent with the predictions of neutral theory. However, another study found that population size and genetic drift were not major factors affecting molecular variation [32], so further research is needed to illustrate the relationship between population size and genetic diversity.

In addition, 200 individuals in this study reached a basic requirement of the effective population [33]; therefore, the first introduced 200 individuals did not cause significant founder effect with the *Ne* value of 3.269 ± 0.968, even after 90 years of independent evolution, the genetic diversity of population is still in the average level.

### 4.4. Phylogenetic Relationship among Populations

Based on 14 haplotypes of mitochondrial genes, the phylogenetic NJ tree and ML rootless tree showed no obvious lineage structure between the Beijing and Shandong populations (Figure 1), so the high genetic differentiation coefficient (*F*st = 0.124) might be due to there being fewer haplotypes in the Beijing population. However, there exists a slight genetic differentiation based on the analysis of microsatellite markers. According to the Bayesian analysis of the Structure software, the *B. orientalis* individuals were divided into two clusters. This genetic discrepancy trend was supported by the differentiation coefficient based on microsatellite markers (*F*st = 0.132, *p* < 0.01), showing the two populations that appeared with significant genetic differentiation. This result indicated that the Beijing population has shown an independent evolutionary trend.

## 5. Conclusions

Although the genetic diversity of the *B. orientalis* population in Beijing was lower than that in Shandong Province, it is still at a normal level which could maintain the survival of the introduced population. A significant genetic differentiation has been formed between the two populations, as the geographic distance between the two populations is great and there are no chances of natural gene exchange between the two populations; therefore, we suggest that the Beijing population should be regarded as a separate management unit, and currently there is no need to reintroduce new animals to improve the genetic diversity of the populations in the Beijing area. This is due to the fact that aimless reintroduction can affect the genetic structure of the local population and lead to genetic drawbacks through crossbreeding. Originating from a small artificially introduced population, the *B. orientalis* in the Beijing region have maintained a moderate genetic diversity after 90 years of independent evolution, indicating that this species has a high capacity for survival and adaptation.

## Figures and Tables

**Figure 1 animals-11-01429-f001:**
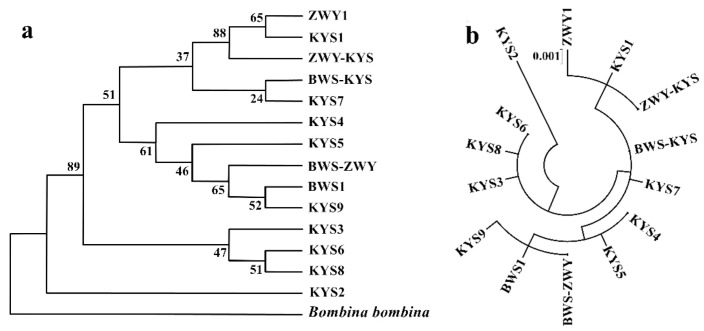
Phylogenetic tree based on mtDNA D-loop region haplotypes. (**a**): Neighbor joining tree. (**b**): Maximum likelihood tree. Confidence degrees on the node were calculated by 1000 bootstraps, and only figures greater than 50% are shown.

**Figure 2 animals-11-01429-f002:**
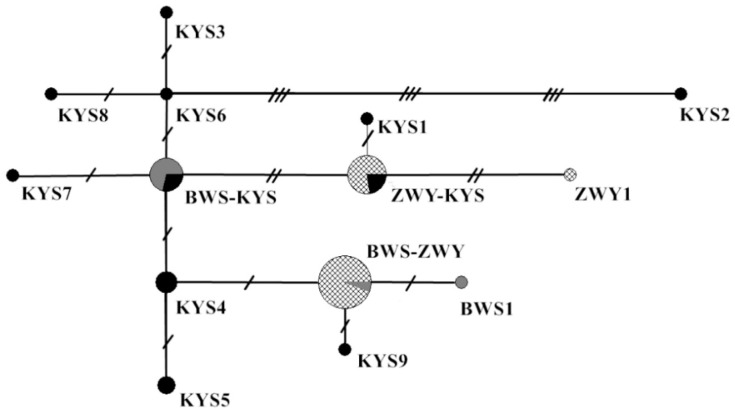
Network diagram based on mtDNA D-loop region haplotypes. The size of circles is related to the frequency of the haplotypes; different patterns in circles represent different populations and short lines represent mutation steps.

**Figure 3 animals-11-01429-f003:**
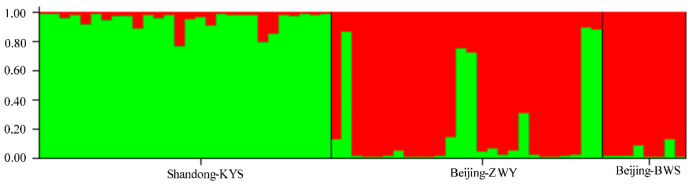
Structure analysis based on microsatellite genotypes among three *Bombina orientalis* populations.

**Table 1 animals-11-01429-t001:** Genetic diversity of *Bombina orientalis* populations based on mitochondrial DNA.

Population	*N*	s	n	h	π	k
Beijing	31	7	5	0.641 ± 0.071	0.00178	1.991
ZWY	24	6	3	0.489 ± 0.084	0.00181	2.022
BWS	7	3	3	0.524 ± 0.209	0.00111	1.238
KYS	16	20	11	0.950 ± 0.036	0.00327	3.658
Total	47	23	14	0.818 ± 0.052	0.00241	2.701

*N*: sample size; s: nucleotide polymorphism sites; n: haplotypes; h: haplotype diversity with SD; π: nucleotide diversity with SD; k: nucleotide difference index.

**Table 2 animals-11-01429-t002:** Genetic diversity of *Bombina orientalis* populations based on microsatellite markers.

Population	*N*a	*N*e	*A*r	*PIC*	*H*o	*H*e	*F*	*PA*
Beijing	7.500 ± 1.176	3.269 ± 0.968	9.243	0.543	0.529 ± 0.114	0.571 ± 0.099	0.090 ± 0.073	14
Shandong	9.667 ± 1.667	6.595 ± 1.786	9.553	0.727	0.726 ± 0.112	0.749 ± 0.081	0.066 ± 0.085	27
Total	27.000 ± 0.302	7.083 ± 1.515	4.557	0.685	0.549 ± 0.078	0.624 ± 0.080	0.112 ± 0.062	-

*N*a: the average allele number with SD; *N*e: the number of effective alleles with SD; *Ar*: the allelic richness; *PIC*: polymorphic information content; *H*o: observed heterozygosity with SD; *H*e: expected heterozygosity with SD; *F*: genetic distance with SD; *PA*: alleles endemic.

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
