# Peer review of "Population Genetic Structure Analysis Reveals Decreased but Moderate Diversity for the Oriental Fire-Bellied Toad Introduced to Beijing after 90 Years of Independent Evolution"

_animals, 2021, doi:10.3390/ani11051429_

Round 1
Reviewer 1 Report
The manuscript entitled "Population genetic structure analysis reveals decreased but moderate diversity for the Oriental fire-bellied toad introduced to Beijing after 90 years of independent evolution" by Yang et al. is aim to characterize two populations of Oriental fire-bellied toads with a complete population genetics toolkit. The authors have a unique opportunity to compare a source population with a completely known background and a population founded 90 years ago after an artificial introduction to the city of Beijing. I congratulate to the authors for the detailed and adequate usage of the methods and the visualization of the results.
However, I have to argue with the applied methods for collecting molecular material. It wasn't an option to use a non-invasive (swabbing) or at least non-lethal sampling (e.g., toe-clips) for collecting DNA samples? If animals were necessary to kill is there any museum or private collection where the carcasses are accessible for further investigation? If there are collection numbers please include it to the text.
Furthermore, despite the fact that this is a well-constructed study with clear findings and nice interpretation of the results, why it is interesting for any other than fire-bellied toad people? I really miss a part at least from the end of manuscript which discusses conservation opportunities of the species and how population genetics can contribute to this. Thus, why the results are interesting for evolutionary biologist?
If the studied populations are inside the native range of B. orientalis why the authors thinks it is necessary (or not) to reintroduce new individuals from the source population? What conservation management issues are suggesting this?
Finally, I have a few minor things for correction:
Line 19: Bombina orientalis should be in italic.
Lines 45, 50, 116...etc. In the whole text Bombina variegate is misspelled. The valid name is Bombina variegata. Change everywhere where it is appears wrongly.
Table 1. It would be nice to give some explanation about column headers in the table legend. Moreover, in “column h” indicate the numbers as SD following the exact value. The same is true for Table 2. Indicate the meanings of the abbreviations.
Lines 236-237: I suppose the brackets are containing the 95% CI, please indicate it in the text.
Line 264: What is "normal level"? Add more discussion on this one.
Author Response
Thank you for your valuable suggestions on revision, It has been revised in the manuscript, Please see the attachment.

Reviewer 2 Report
The manuscript presents interesting research for the conservation and understanding of evolutionary biology of a species of amphibian. Suitable and sufficient molecular tools are employed effectively to assess the questions presented by the authors.
Besides minor suggestions (shown in the attached manuscript) the manuscript is suitable for publishing.

Author Response

(The authors gave the same response as above.)
